# GC Insights: "Sedimentary Rock!" – A web app that converts geological strata data into music

Toshiyuki Kitazawa[1], Kazuaki Aoki[1], Hiroaki Yasutomo[2], Takuma Kato[2]

[1]Faculty of Geo-Environmental Science, Rissho University, Kumagaya, Saitama, 360-0194, Japan
[2]SHIFTBRAIN Inc., Setagaya, Tokyo, 155-0033, Japan

*Correspondence to*: Toshiyuki Kitazawa (kitazawa@ris.ac.jp)

**Abstract.** A web app called "Sedimentary Rock!" (https://rissho-es.jp/kitazawa/SedimentaryRock/) has been developed to reveal geological strata to human hearing and rhythmic senses, much like music, in a straightforward manner. The web app was designed and opened for non-specialists in geology. A columnar section created on the app is converted to notes and played. The thickness of a layer is converted to the length of a note, and the grain size is converted to pitch level. We converted schematic columnar sections of several environments and real Pleistocene strata from Vietnam into music and surveyed the educational effect of the web app using a questionnaire. Sedimentary Rock! makes learners understand that the strata consist of concepts of grain size and thickness of stratum, and provides a new intuitive experience to listen to geological history.

## 1 Introduction

Geological strata were formed by the upward stacking of sediments and record the geological history, the changes in time of the local or global environment. The most essential feature of strata is grain size, which is divided into gravel, sand, and mud with more detailed subdivisions, indicating the strength of the current that caused the deposition. Another is the thickness of the bed, which means water depth, sediment discharge, time length, *etc*. Primary information obtained from geological strata is generally interpreted visually. However, methods other than visual inspection can be used to interpret geological strata, such as verifying grain size by touch with the fingers and through physical surveys where strata cannot be directly observed underground. New techniques and perspectives can interpret strata, which may lead to discoveries and understanding. Our motivation for this study is to convert researchers' and educators' field data into music, and learners can create music from geological records.

Attempts to make natural phenomena audible that we know as "sonification" converted atom trap (Sturm, 2001), brainwaves (Sonifyer.org, 2006), the behavior and structure of molecules (Nakamura, 2012; Mahjour et al., 2023), colors (Harbisson, 2012), and molecular evolution (Martin et al., 2024) into sounds. In the geoscience field, some attempts at "musification" have been made, which involve creating music from geoscience data. The sonification of seismic signals for 2 days was presented in a video 10 days after the 2011 Tohoku Earthquake (Sonifyer.org, 2011). Volcanic geophysical information on seismic movements was collected for sound analysis in eruption dynamics that can be used to predict the future activity of a volcano (Avanzo et al., 2012). Climate data were transformed into acoustic signals, and videos playing the music were distributed to enhance the communication (Planetary Bands, 2015; St. George et al., 2017). By using simulated data instead of observational data,

the climate model was musificated from time periods outside the recent past and possible future (de Mora et al., 2020). Observed energy budget data were converted into sound and aligned more closely
with traditional music (Nagai, 2024). Terrain relief was translated into acoustic representations for landform quantitative analysis in drainage basins (Lin et al., 2024). These attempts were made by geoscience specialists, and the performance or video of sound was shared with the audience, including non-specialists.

To reveal geological strata to human hearing and rhythmic senses, we have developed a web app,
Sedimentary Rock! (Kitazawa et al., 2017), which plays music by converting strata thicknesses to note lengths and grain sizes to pitch levels. The access has been approximately 1,000 since we opened the web app in 2017. The web app was developed and made available not only to specialists in geology but also to non-specialists. Musification may help non-specialists to understand the cyclicity and drastic changes in geological history recorded in strata, as well as the variation in each sedimentary
environment. For example, an upward-fining succession forms during a coastal channel-fill process, and the stacking of such successions represents that several channels formed and filled in a stable environment. Then, if the sea level fell, these strata would be eroded and overlain by coarser terrestrial deposits. If the rhythmic and cyclic melody, converted from upward-fining successions, abruptly changes to a significantly different melody, the drastic change may be recognized as unconformity. We
also surveyed the educational effect of the web app using a questionnaire.

## 2  Sedimentary Rock!

The web app was created with JavaScript and works in web browsers for computers, smartphones, and tablets (Fig. 1a). The procedure when using the web app is to create and edit a single columnar section
in the web app and then convert it to music. The web app may create columnar sections without playing music, and descriptions and remarks can be written in a memo box. The user must initially connect to the Rissho University server (Kitazawa et al., 2017), but once the web app has been started up, columnar sections can be created, saved, and loaded, and music can be played, even offline, for as long as the browser window is kept open. The web app will be helpful in education and outreach activities
because it is easy for anyone to use freely.

## 3  Creating columnar sections

A columnar section is a geologic illustration that shows a vertical stack of rock types; vertical width represents the thickness, and horizontal width often indicates the grain size. Using the web app, a
columnar section is created by entering a single layer's thickness and grain size and adding more layers above or below. After the columnar section is created, layers can be freely selected and edited, and further layers can be inserted at selected positions. Thicknesses can be entered in units of m, cm, or mm, ranging from 1 mm to 100 m. In the rare case of a layer with a thickness of more than 100 m, multiple 100-m sections can be individually added. Grain sizes are entered as selections from eight types—
"gravel", "very coarse sand", "coarse sand", "medium sand", "fine sand", "very fine sand", "silt", and "clay"—or "none" (representing a gap or core damage). In a created columnar section, gravel is colored brown, sands are colored yellow, and mud is colored blue. The grain size can be varied between the bottom and top of a layer, which are then interpolated linearly. Thus, grading and reverse grading can be represented. Most grain size variations can be represented by entering separate positions where

variation trends change. Other functions include adjusting a columnar section's horizontal width and length to match the screen size or changing its visual impact. Black boundary lines can be displayed at bedding planes, although the default setting does not display these lines.

When information on the thicknesses and grain sizes of a created columnar section is saved, the information is exported in the "Sedimentary Rock Data (SRD)" format that we developed, which is an
image of a pattern of colored squares. SRD is an ordinary image file (PNG), which can be saved in an image folder on a smartphone and sent to other people. A saved columnar section can be loaded into the web app by selecting the corresponding SRD file. A loaded columnar section can continue to be edited and, if saved again, saved as a new SRD file rather than overwriting the old one. As a result, Sedimentary Rock! is not only a musification app but also a helpful tool for making and presenting
columnar sections and managing editable sedimentological data.

## 4 Converting strata to music

For the conversion of the thickness t (mm) of a layer to the length l (ms) of a note, it is specified that the thicker the layer, the longer the note. In order to handle strata with a wide range of thicknesses, the
natural logarithm of the thickness is used, $l = 100\log t$. The thickness of each layer can be from 1 mm to 100 m, corresponding to note lengths from 0 to 1151 ms. The A minor pentatonic scale is used to convert from grain size to pitch level. It tends not to produce unpleasant sounds because it has no semitone intervals. It is a basic scale found in traditional music across parts of the world and is often used in rock music today. Eight tones starting from A at 440 Hz are specified such that the smaller the
grain size, the higher the pitch. That is, gravel is A, very coarse sand is C, coarse sand is D, medium sand is E, fine sand is G, very fine sand is high A, silt is high C, and clay is high D. "None" is represented by silence. A graded bed or inverse-graded bed is expressed by glissando (slides) from one pitch to another.

When the Play button is pressed, the columnar section, starting from the bottom, is converted to notes
and played. This sequence of sounds can be listened to as music (whether it actually "rocks" is another matter), and the strata can be heard. When the strata are not thick and the columnar section is short, the output sound sequence often does not form a self-contained musical composition. The sound may be better understood as a "musical motif", a component within a larger musical piece. The motif may reflect a general or cyclic pattern of environmental change in the longer-term history of the Earth. The
music would need improvement to sound pleasant, but the basic objective of revealing geological strata to human hearing and rhythmic senses is achieved.

We shared a video on how to use Sedimentary Rocks! and several examples of musification from schematic columnar sections of typical depositional environments of the delta, wave-dominated coast, tidal flat, and submarine fan (Kitazawa, 2024) (https://www.youtube.com/@SedimentaryRock-
e7r/videos).

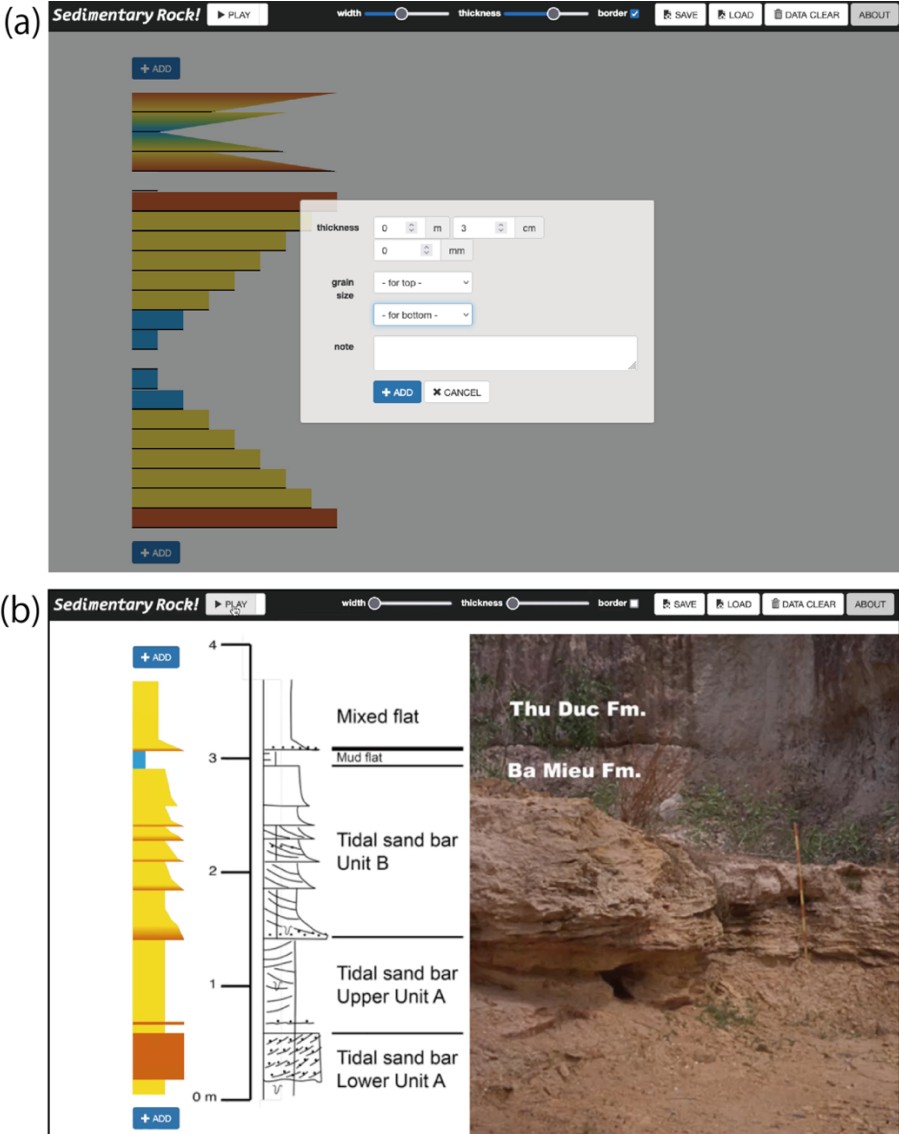

**Figure 1. (a)** Interface of "Sedimentary Rock!". A columnar section is created and edited by entering a single layer's thickness and grain size. **(b)** Musification from the Pleistocene tidal deposits. A columnar section on the left was created on Sedimentary Rock!, and another was modified from Kitazawa and Tateishi (2005).

## 5  Musification of geological history

A case study of musification from a columnar section of tidal deposits was done (Fig. 1b). The geological strata data were collected from the Pleistocene Ba Mieu and Thu Duc Formations (about 50,000-200,000 years ago) in southern Vietnam (Kitazawa et al., 2006). The Ba Mieu Formation on this outcrop is a tide-dominated delta succession of the paleo Dong Nai River during the highstand of

marine isotope stage (MIS) 5 (Kitazawa, 2007), indicating stacks of two tidal sand bars and being buried by muddy tidal flat deposits (Kitazawa and Tateishi, 2005). Upward convex bar-shaped
sediments of Units A and B consist of cross-bedded coarse-grained sediments transported by ebb tidal currents and mud layers (mud drapes) which formed in slack water during high or low tide. The tidal range estimated from around deposits is about 5 m. The music converted from the tidal sand bar deposits is a cyclic melody indicating upward-fining successions due to frequent depositional cycles of development and migration of tidal sand bars. The delta progradation resulted in the shallowing,
weakening of tidal current, and deposition of the fine mud in the mud flat. The longer and larger environmental change is converted to a gradually higher tone in the music.

   The mud flat deposits are unconformably overlain by mixed flat deposits of the Thu Duc Formation, and the unconformity is amalgamated with a wave/tidal ravinement surface (W/TRS) that is typical in the seaward part of the paleo Dong Nai estuary during a transgression of MIS 5 (Kitazawa, 2007;
Kitazawa and Murakoshi, 2016). Thus, there is a remarkable erosion and a loss of previously formed records between the Ba Mieu and Thu Duc Formations. The drastic change in geological history is converted into an abrupt melody change.

## 6  Educational effects and potential outreach

We surveyed the educational effects for improving learner's understanding of strata and potential outreach of Sedimentary Rock! with a questionnaire. The 29 answers included 21 university students who started to study geology, and 3 of them were learning in a science teacher licensee course, 1 science teacher, 2 educators in museums, and 5 others. At first, we provided participants with an explanation about the function of "Sedimentary Rock!" and the concepts of grain size and columnar
section. Next, they used Sedimentary Rock!, made a columnar section, listened to several examples of musification from schematic columnar sections of typical depositional environments, and then answered four questions on a 1-5 scale (Table A1).

   Most participants strongly agreed (76% strongly agree, 21% agree) that creating a columnar section deepened their understanding of the concepts of grain size and thickness of the stratum. The function of
creating a columnar section in Sedimentary Rock! was useful for improving learner's visual understanding. They also agreed (45% strongly agree, 41% agree) that adding hearing and rhythmic senses, as well as visual senses such as outcrop pictures or columnar sections, was useful for understanding strata. Although it was less than visual information from a columnar section, hearing information also had certain effects. The melody itself, output from Sedimentary Rock!, had certain
effects (41% strongly agree, 34% agree) to improve understanding that the stacks of strata represent the geological history, such as depositional environments. Changes in the melody had a greater impact (48% strongly agree, 31% agree) on improving knowledge that grain size and thickness changes indicate environmental changes. Those results suggest that musification in the geology field has educational effects, and Sedimentary Rock! is a useful tool for connecting visual information from a
columnar section with auditory information, deepening understanding of the relation between strata and environment.

   Although the concept of Sedimentary Rock! is to reveal geological strata to human hearing, visual inspection remains the most important way to interpret strata. For university students studying geology, the ideal way to use Sedimentary Rock! is to transcribe their own columnar sections or descriptions

from fieldwork. Converting their own columnar sections into music would lead to new discoveries and a deeper understanding, for example, of the general or cyclic patterns of environmental changes and drastic changes in geological history.

More effort, however, it may be required of school-age students to create their own columnar section. Teachers may need to prepare columnar sections, save them as SRD files, and share them with the students. Even in that case, an experience to make or transcribe a columnar section by themselves is effective in understanding the most essential feature of strata: grain size and thickness. When students listen first, geological music may stimulate their curiosity before formal instruction. It may also be interesting that well-known music is resolved into elements of note length and pitch level, and converted into a columnar section: strata.

In addition, Sedimentary Rock! has potential applications not only in education but also in outreach and understanding of geology and sedimentology for non-specialists. We also consider the potential benefits to users with disabilities. Although we have never had the opportunity to survey, people with visual impairments could interpret geological strata using the sound of Sedimentary Rock! Thus, we should continue to update the Sedimentary Rock! to enhance operations and sounding. There is room for improvement in the web app interface in the future, for example, by adding thickness and color scales, and making it easier to edit layers. On the musical side, we can also improve the timbre and tempo.

## 7  Conclusions

Sedimentary Rock! is a tool that converts columnar sections into human hearing and rhythmic senses. It is easy for anyone to use, and educators can make learners understand that the strata consist of concepts of grain size and thickness of the stratum by working to make a columnar section. Furthermore, the learners understand that the stacks of strata represent the geological history, and the changes in grain size and thickness indicate the environmental changes not only with visual senses, such as outcrop pictures or columnar sections, but also in hearing and rhythmic senses. Sedimentary Rock! provides a new, intuitive experience to listen to the music of geological history.

**Supplement link.** The web app "Sedimentary Rock!" is available at: http://rissho-es.jp/kitazawa/SedimentaryRock/

The code is available at: https://github.com/kazu404/SedimentaryRock.git

**Author contribution.** All authors contributed to the design of the concepts of the web app "Sedimentary Rock!". HiroakiY and TakumaK developed the code and created the Sedimentary Rock Data (SRD) format. KazuakiA administers the web app. ToshiyukiK collected and interpreted geological data. ToshiyukiK prepared the manuscript with contributions from all co-authors.

**Competing interests.** The authors declare that they have no conflict of interest.

**Ethical statement.** The questionnaire in this article have been reviewed by the Ethics Committee, Institute of Environmental Science, Rissho University, and cleared of ethical problems.

**Disclaimer.** This article was written and published on volunteered time by the authors.

**Acknowledgements.** We thank Takuma Noda for his basic study on this paper. We thank cooperators
for the questionnaire.

**Review statement.** This paper was edited by Sebastian G. Mutz and reviewed by Lee de Mora and two
anonymous referees.

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
