# Peer review of "GC Insights: “Sedimentary Rock!” – A web app that converts geological strata data into music"

_EGUsphere, 2025_

## Author Response (AR1)

egusphere-2025-1978
**Title: GC Insights: "Sedimentary Rock!" – A web app that converts geological strata data into music**
**Author(s): Toshiyuki Kitazawa et al.**

**Author's response to the Referee #1**

Thank you for accepting the role of referee and for your meaningful suggestions. Our response to the referee #1 comments is as follows.

1) In the introduction, mention of previous attempts of sonification were made (Lines 21-29). I think it would be of value for the authors to expound a little more on these studies. Specifically, why was sound added, what was the intended impact, how was it evaluated, and were those previous studies successful? This may help frame the value and significance of this particular program.
> We revised the paper in consideration of the referee comments.

2) The potential benefits to users with disabilities should be also discussed in this paper. Specifically, could the program be adapted so people with visual impairments can interpret geological strata?
> Although there was such a discussion among the authors before and some comments in the questionnaire, we have never had the opportunity to survey the benefits to users with disabilities. We revised "6 Educational effects" in consideration of the referee comments.

3) Is it possible to modify the program and slow the play rate? (E.g. a scale bar from 0-100% speed). For me, at least, it plays too fast for me to interpret the trends in my head (which would be a barrier to the effectiveness and usefulness of listening to the strata.
> We appreciate the referee suggestions, which are useful for proceeding with our app. However, we would like to avoid the contradictions in the paper between the web app version used in the questionnaire and the one we will update. We have also received further improvements from another referee; thus, we will take some time to update the app. For example, playing slow creates a problem where the sound from much thicker strata becomes too long; thus, the adjustable play rate is a very good idea, but requires a higher technical level. We prefer to publish the current version at this stage and then carefully update the app, taking into account the referee comments.

4) Relating to the survey and demographics, were there any trends seen that could be discussed? For example, how did the students compare to the three educators (museum and teacher), or those in education vs the 5 others?
> We couldn't find trends between the attributes.

Additional comments:
Line 32: "The web app was opened in 2018 and has been used." This statement is very vague; how many people have used or engaged with the software?
> The access to the web app has been approximately 1000 since 2018. We mentioned it.

Figure caption '(2)' should be replaced with '(b)'
> We revised according to the referee comment.

Line 114: relace 'a science teacher' with '1 science teacher'
> We revised according to the referee comment.

Line 125: as the software has been online for 7 years now (since 2018), consider removing the word 'new'
> We revised according to the referee comment.

egusphere-2025-1978
Title: GC Insights: "Sedimentary Rock!" – A web app that converts geological strata data into music
Author(s): Toshiyuki Kitazawa et al.

**Author's response to the Referee #2**

Thank you for accepting the role of referee and for your meaningful suggestions. Our response to Dr. Lee de Mora (the referee #2) comments is as follows.

My primary criticism is that the paper and the web app do not go far enough to explain sedimentary rocks. This web app and manuscript should be of interest beyond just the geological community. As such, it would really be worth spending some time explaining some of the core concepts and why they are interesting and exciting: ie, what are geological strata? what is a columnar section? Why are they interesting? What can they tell us about the local geological history?
> We revised the paper in consideration of the referee comments.

In either case, I would say that this is an interesting paper, and should be published with major corrections. I do not want to disparage the hard work of the authors,  the concept is great, and with a little work, the rest of the paper could be really great too.  I think that a small number of hopefully simple changes to paper and the app, like a depth scale along the y axis, a colour legend, a few presets with explanations would go a long way in improving it.
> We appreciate the referee suggestions, which are useful for proceeding with our app. However, we would like to avoid the contradictions in the paper between the web app version used in the questionnaire and the one we will update. We have also received further improvements from another referee; thus, we will take some time to update the app. We prefer to publish the current version at this stage and then carefully update the app, taking into account the referee comments.

The example showing the app to demonstrate different rock types in section 5 is great. I'd love to see more examples of this in the paper.
> The concept of the app is to allow learners to create sounds by themselves, so no more examples are needed, we think.

I don't think that the survey in section 6 and table 1 is very insightful
> The questionnaire survey was added according to the executive editor's request when the first submission of this paper was rejected. The questionnaire was needed

to indicate that the app is useful in education; educators can make learners understand that the strata consist of concepts of grain size and thickness of the stratum by working to make a columnar section.

While my role here is to review the paper, not the app, I have a few comments on the app:
The initial start-up pattern is an interesting example, but I find the examples on your youtube channel to be a lot more engaging. In addition, the effort needed to create your own columnal section is a bit high. I can't imagine many casual users doing this. However, lots of interesting datasets have been shown via youtube. It would be really nice to be able to explore the datasets shown on youtube from the web app. This could be done as a dropdown list of preset starting conditions, or buttons (as in, several different datasets to explore instead of just the standard pattern).
> We prefer to update after publication.

Similarly, if the data is based on a real world location, an explanations about the different kinds of rock, where that occurs,  and why it's interesting would make it a much more powerful education tool.
> The examples are schematic ones I create, thus not real.

There is no depth/thickness information on the web app. A y-axis scale would allow a much better interpretation of what we're looking at.
> We prefer to update after publication.

Also, there is no legend in the webapp. This is really needed to help understand what kind of rock we're looking at.
All this information is shown in Figure 1b), so the authors are aware that this is needed to explain the strata, but figure 1a) is closer to what the app looks like, and is much harder to interpret.  I would like the app to look more like figure 1b): includes a depth scale, a colour scale, and some notes to help explain what we're seeing and hearing.
> We prefer to update after publication.

Finally, what data is shown on the x-axis? It seems to be linked to pitch and grain size? Is a bar chart the most appropriate way to show this? Would textures as well as colours make more visual sense that colours?
> X-axis indicates the grain size. It is a general method to create a columnar section, and we explained what a columnar section is at the beginning of "3

Creating columnar sections". Texture is a good idea, but it requires a higher technical level, because the gradual change, for example, from sand to mud, is difficult to show. So, we prefer to consider after publication.

The webapp interface could be improved. For instance, click and hold to drag layers around with the mouse and rearrange them. Right click to edit a layer.
> We prefer to update after publication.

On the musical side, the tone is pleasant enough, and pitch is a good way to link to information. However, I always feel that tone and timbre would make the audio more exciting? I always recommend checking Flowers (2005) for a great summary of audiolisations, which may inspire new directions for your work.
Flowers, J. H.: Thirteen years of reflection on auditory graphing: promises, pitfalls and potential new directions, Proceedings of ICAD 05-Eleventh Meeting of the International Conference on Auditory Display, Limerick, Ireland, 6–9 July, 406–409, 2005
It seems like the longest that a piece of music can be is 5-20 seconds? Is it possible to change the tempo? Or create a loop? This may make it more musical. The youtube videos all show the pattern repeating, so adding a way to do that would be useful. The other issue is that music requires a regular pulse and regular repeating patterns, but in this work, the note duration is linked to thickness, so it may not invoke a sense of rhythm.
> It is a challenging issue that we must choose between a straightforward scientific approach and an engaging artistic approach in sonification. Furthermore, a technical issue exists. Regarding the tempo, for example, playing slow creates a problem where the sound from much thicker strata becomes too long; thus, the adjustable play rate is a very good idea, but requires a higher technical level. So, we prefer to take some time.

Specific comments:
L15: Please add a sentence to this paragraph about why we should be interested in sedimentary layers of rock. What we can learn from it or use it for.
> We revised the paper in consideration of the referee comments.

L15: What is strata?
> We revised according to the referee comment.

L16: What is grain size?
> We revised according to the referee comment.

L25-L29: Single sentence paragraph – merge with previous paragraph. Also, remove "Especially".
> We revised according to the referee comment.

L30: remove "very simply"
> We revised according to the referee comment.

L32: "The web app was opened in 2018 and has been used." While this is true, I suspect that this sentence is missing some words.
> We revised the paper in consideration of the referee comments and mention that access to the web app has been approximately 1000 since 2018.

L32: remove "anyone"
> We revised according to the referee comment.

L38: remove general
> We revised according to the referee comment.

L41: Should this be a reference, instead of a web address in line?
> We revised according to the referee comment.

L43: Don't start a sentence with Because. (Because is a conjunction word, so it's only just to join two sentence parts).
> We revised according to the referee comment.

L41-44: Actually, this is not that useful to know. You can probably delete "but once the web app has been started up, columnar sections can be created, saved, and loaded, and music can be played, even offline, for as long as the browser window is kept open."
> We believe it is important information for us, fieldworkers. We often survey and make columnar sections far from the internet.

L44: Is the code open source and hosted online anywhere? Can users download it and make their own server? There's a link to the github in the supplementatal code section, but maybe mention it here too.
> We currently allow use from our server because we would like to analyze the access and manage updates.

L46: What is a columnal Section?

> We revised according to the referee comment.

L53: This is the legend and should appear in the web app and in figure 1.
> We prefer to update after publication.

L66: Single sentence paragraph, merge with previous
> We revised according to the referee comment.

L72: l=100logt: write this as an in line equation.
> We revised according to the referee comment.

L73: Note duration in ms is not a musical way to describe this. Have you considered another way to link thickness to tempo? Perhaps binning thicknesses into musical notes? 100m = 2 bars, 1m = crochet, 1mm = semiquaver or something scaling like this?
> We prefer to consider after publication. It is also the issue that we must choose between a straightforward scientific approach and an engaging artistic approach in sonification.

L77: If you're starting the scale from A, then it's Aminor pentanic, not C major.
> We revised according to the referee comment.

L77-78: You have 8 types of grain size, why use a pentatonic scale instead of an 8 note major scale?
> We prefer to consider after publication. It is also the issue that we must choose between a scientific and an artistic approach. Currently, we have chosen a straightforward scientific approach because the A minor pentatonic scale has no semitone intervals, and it corresponds to the classification of grain size defined by regular intervals of logarithms.

L78: One of the really unique elements of your sonification is the glissando (slides) from one pitch to another, but you don't mention this in the text. It would be worth adding a sentence describing this.
> Thank you. We revised according to the referee comment.

L104-109: I really like this, this is what the tool should do! Help understand the layers!
> Thank you. We added explanations in "5 Musification of geological history" according to the referee comment.

L111-119: I am not sure about this survey. It doesn't appear in the abstract or the methods. I understand the intention, but I feel that these may be leading questions, and I'm not sure that it adds very much to the overall message of the paper.
> We revised the abstract and introduction in consideration of the referee comments.

I'd like to see a Discussions, limitations and extensions section. What are the issues with this approach? Where can it be extended in the future?
> We revised "**6 Educational effects**" in consideration of the referee comments, including the ideas of updating the app and the potential benefits to users with disabilities.

In the supplemental code github repository, you should add instructions on how to install and run the app to the README.md file.
> We currently allow use from our server because we would like to analyze the access and manage updates.